# Adaptive Retrofit for Adaptive Reuse: Converting an Industrial Chimney into a Ventilation Duct to Improve Internal Comfort in a Historic Environment

**Mariangela De Vita [1,*]** , **Francesco Duronio [2]** , **Angelo De Vita [2]** and **Pierluigi De Berardinis [1]**

[1] Department of Civil, Building and Environmental Engineering, University of L'Aquila, 67100 L'Aquila, Italy; pierluigi.deberardinis@univaq.it

[2] Department of Industrial and Information Engineering and Economics, University of L'Aquila, 67100 L'Aquila, Italy; francesco.duronio@univaq.it (F.D.); angelo.devita@univaq.it (A.D.V.)

* Correspondence: ing.mariangeladevita@gmail.com

**Abstract:** The reuse of architectural heritage is a topic of great interest for scientific research, involving aspects ranging from the architectural compatibility of the interventions to the performance updating of the artefacts, from the point of view of both energy consumption and internal comfort suitable for the new use. Compatible technological solutions exploit the passive cooling activating latent physical mechanisms of the building, of the envelope or its parts, such as openings and disused shafts. This work concerns the conversion of an old chimney, completely integrated into the historical envelope, into a ventilation duct for the air exchange and the internal comfort improvement of an old factory, proposing an adaptive retrofit solution during adaptive reuse intervention. Thermo-fluid dynamics analyses, performed with an ad hoc CFD solver for flows with flotation effects, verified the effective functionality of the device in summer and winter conditions. The results show that, in summer, the activation of passive ventilation improves the indoor comfort of the environment, while, in winter, it worsens them. This study demonstrates the usefulness of activating passive cooling phenomena in preserving historical architecture. Finally, the future potential of the application is presented by integrating the ventilation chimney with a mechanical control system to optimize its operation even in winter conditions.

**Keywords:** indoor comfort; ventilation duct; industrial heritage; adaptive retrofit; CFD analyses

## 1. Introduction

Over the previous decades, national and international governments defined the measures and standards to lead the construction sector to meet the requirements of reducing pollutants and saving energy [1]. In the E.U., the performance targets are periodically updated by issuing new directives [2,3] to achieve climate protection [4]; these targets are implemented at the national level through implementing rules of the aforementioned directives.

In the context of environmental sustainability and energy saving, the construction sector has always been particularly attentive, as it is responsible for a significant energy expenditure [5–7] along the entire production chain, asset management and disposal at the end of life. In this scenario, the historical built heritage represents an important resource, consisting of many buildings subject to protection that are in a state of deterioration, neglect or that show significant deficits in environmental performance [8–11]. Therefore, the architectural heritage requires interventions that look at conservation and, simultaneously, at performance enhancement and updating to greatly contribute to building a sustainable society, as the Baukultur theories show [12–14].

In recent years, the focus on retrofitting historic buildings to better their environmental performance has increasingly taken on a character of enhancement using reversible, temporary and low material impact solutions on structures [12,13] at the expense, finally, of more

invasive solutions that are hyper-technological but, at the same time, that are capable of distorting the natural environmental behaviour of the building. In fact, the overuse of heating, ventilation, and air conditioning systems (HVAC) and, with them, the hyper-technology applied to the heritage has often had enormous repercussions on the internal microclimate of the restored buildings, causing irreversible damage to the elements and structures and compromising the original facies and its spatial and decorative values [15,16].

The most recent methodologies on architectural heritage reuse are instead based on adaptivity, which can be understood as a 2.0 version of resilience. Making the architectural heritage adaptive is equivalent to making the conservation and enhancement intervention not utter and stark but open to future changes—if the surrounding conditions such as fruition, use and performance updating should modify over time—to be carried out easily, often by users themselves, without causing damage to structures and components [17–21].

Reading the adaptive reuse under the lens of environmental sustainability represents a fundamental step to ensure that the energy and structural retrofit interventions align with the new guidelines for renovating historic buildings.

Talking about adaptive retrofit declares the intent to collect all the forward steps taken by contemporary reuse and restoration theories and convert them into architectural technology. Adaptive retrofit interventions can guarantee complete reversibility, the respect and protection of historical and architectural values, flexibility of use and, at the same time, a quantitatively predictable performance update of the entire buildings involved during the design phase that is measurable on-site [21].

The performance improvement of the architectural heritage can be pursued following two strategies to be applied alternately or simultaneously (hybrid solutions): active control strategies (through interventions on plant systems) and passive control strategies (through interventions aimed at enhancing the natural environmental behaviour of the building) [16]. It has been shown that, although the active control strategy is often necessary to compensate for the performance deficits of the historical structure, it risks compromising its statics (when new systems have too strong of an impact on the structures) and architectural values [15,16,22]. On the contrary, passive control is the first step in compliance with conservation principles, in line with the dictates of restoration on protection issues [23–26]. Ordinarily, passive strategies are applied through interventions on the envelope (opaque and/or transparent) to improve its thermal transmittance or to solve thermal bridges, interstitial condensation, and capillary rising phenomena [16].

Further technological solutions, which are less applied but high performance, to increase the comfort of buildings—especially in a hot, humid and temperate climate—and, therefore, to limit the use of active systems and, consequently, energy consumption, are the activation of ventilation phenomena for passive cooling for the air exchange. Such a task should be pursued by providing special ventilation chimneys [27–30]. Inserting these devices in the historical envelope often strongly impacts the point of discouraging their application. Therefore, the resolution of integrating ventilation chimneys into the architectural heritage envelope is relevant because of the low material impact on the pre-existing structure and the applicability of adaptive and reversible technological solutions, which would allow easy restoring the ante-operam status in the future.

Many studies have looked at ventilation chimneys through mathematical simulations and experimental investigations: this choice of passive ventilation depends on design parameters and the thermal performances for different geometrical configurations. Research has shown that airspeeds in chimneys are influenced by the channel's width and the angle of inclination of the chimney. Despite a large amount of literature on analytical studies of ventilation chimneys operation, widely validated by CFD analysis and optimized in geometry [31–34], there is a lack of research into integrating these systems in historical buildings.

As described in [35–38], the architectural heritage often has morphological characteristics such as exploiting natural ventilation as a bioclimatic control. The hot climate of Valencia causes discomfort, especially in summer, where passive cooling would bring

benefits. The presence of chimneys to be optimized as ventilation chimneys is often found in industrial buildings; in other cases, there are shed roofs with openings suitable for the purpose. In this work, the problem of integration exposed above, and the comfort issue, are resolved by exploiting a historicized, pre-existing, and disused element of the envelope: an old industrial chimney. This chimney has been converted into a ventilation duct for the passive cooling and air changing of the adjacent factory's big open space. In this way, the device for the environmental performance improvement of the building is naturally integrated into the architectural complex. Therefore, the structure's enhancement is implemented by adopting just light interventions with low material impact on the historic envelope. The historical structure shown in this work is the Aceitera de Marxalenes, an old oil factory located in Valencia and currently under renovation by the city council. The functionality and efficiency of the designed ventilation duct have been validated by using CFD simulations, both in the winter and summer regime. A specific OpenFOAM solver for airflows featuring buoyancy effects has been selected. This approach has been validated against experimental data of a scaled ventilation tower and then applied to the Aceitera case study. OpenFOAM is an open-source code that allows the simulation of various engineering processes, particularly passive and forced convection [39]. The simulation's results allow the evaluation of the ventilation performance within the main room of the building and the thermal comfort achieved after a certain period of air exchange through the chimney. More precisely, exploiting the CFD results, thermal comfort has been evaluated using the predicted mean vote (PMV) approach to estimate the human sensation of thermal comfort. The results shed light on the right strategy for controlling the air-exchange process.

## 2. Materials and Methods

The paper aims to demonstrate the possibility of optimizing the environmental performance of historical artefacts by exploiting construction elements already integrated into the building.

This paper analyzes a case study, the historic chimney of the Aceitera de Marxalenes in Valencia, an ancient oil mill currently undergoing a renovation that will bring the structure to a change of use, becoming a centre for the elderly and a museum (Figure 1).

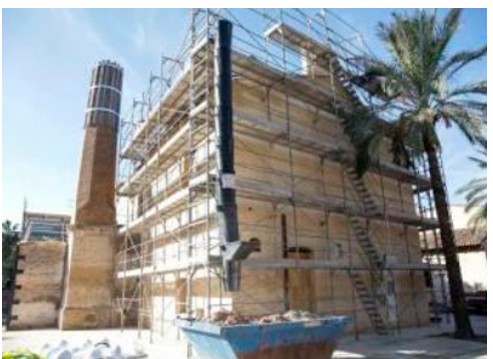 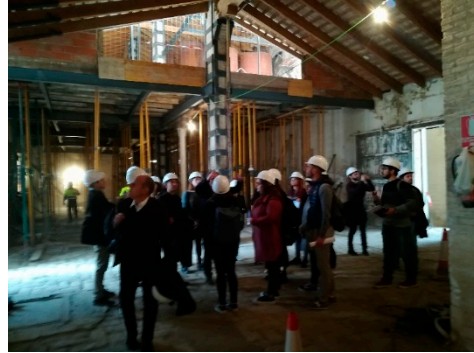

**Figure 1.** The Aceitera de Marxalenes during the restoration works. On the **left**: an external view of the chimney; On the **right**: internal view of the productive open space at ground level.

The analyses carried out on-site reveal an absence of heating and cooling systems in the building, so the internal microclimate and comfort depend exclusively on the external climatic conditions and on the envelope's thermal performance. Although the intervention carried out by the municipal administration does not provide for it, the new intended fruition would, therefore, imply the need for energy improvement interventions to allow the new use of the structure to take place in conditions of thermo-hygrometric comfort and healthiness of the air.

Therefore, this study involved the conversion of the old chimney into a ventilation chimney to activate passive cooling phenomena in the structure to improve internal comfort.

The conversion of the chimney into a ventilation duct was possible thanks to some architectural and technological solutions, designed according to the historical and architectural analysis of the building and validated by CFD simulations.

*2.1. Industrial Heritage Reuse in Valencia*

This work stems from an in-depth analysis of Valencia's industrial architectural heritage (Spain), carried out thanks to the short-term mobility research funding from the Italian National Research Council for the years 2018 and 2019. The analysis made it possible to assess the state of the art of the city's industrial architecture, revealing a considerable heritage, a portion of which has already been restored, with a change of use towards one that is often very far from the original one [40].

The restructuring of the Valencia industrial heritage is currently underway and, according to the analysis carried out, still shows significant unexplored potential. In particular, the interventions carried out to date have focused a lot on the aspect of functional reuse from the perspective of an urban and neighbourhood redevelopment of the intervention areas, often with very positive results, as in the case of Bombas gens and Fabrica de Hielo [21,40]. Furthermore, the environmental issues seem to be forgotten, although the structures involved in the restoration process are suitable for accommodating numerous adaptive retrofit solutions whose experimentation would have low cost, both from an economic investment point of view and consider the material impact on the industrial protected heritage.

In most cases, the factories have exposed brick masonry with thermal capacity adequate for the climatic reference zone (hot-summer Mediterranean climate—Köppen climate classification: Csa); therefore, an improvement in transmittance is not considered necessary. Furthermore, for figurative reasons, covering the wall support with insulating materials— even by using dry and reversible systems—would change the architectural language of the buildings. On the contrary, solutions that introduce devices for cooling with natural or mechanical ventilation would be desirable.

However, it should be noted that most of these structures, as is typical in all Valencian buildings, residential and otherwise, are devoid of heating and air conditioning systems; the main environmental problem that is detected is the relative humidity, ranging from 59% in June to 70% in October, which leads to a condition of discomfort in the buildings, both in summer and in winter [41]. Although the addition of an HVAC system would be decisive for the problem highlighted in several cases, the risk of too strong an impact on the contest must be considered [42]. This is what happened, for example, in the restoration of the *Almudin*, an ancient spectacular barn in Valencia, where the new system installed (underused in operation) compromised the spatial perception of the historic deambulatory [16].

Even the problem of excessive air humidity can, therefore, be effectively counteracted through natural ventilation activation. In buildings such as the ancient Valencian factories, ventilation chimneys or air exchange mechanisms can be easily activated by exploiting the morphological and geometric characteristics of the structures (long *naves* with one or more trusses characterized by a single big environment, saw tooth covering, presence of disused chimneys, presence of underground rooms, etc.).

*2.2. The case Study of the Aceitera de Marxalenes: The Activation of the Ventilation Chimney*

The peculiarity of Valencian industrial architecture depends both on the large number of industrial buildings falling within the city's heritage and on the need to redevelop them due to the widespread deterioration of these structures [40]. Among these, there is the *Aceitera de Marxalenes*, an ancient oil mill currently owned by the municipality and undergoing renovation. The structure is part of what, in the past, was popularly known as the commercial and residential area of the Barrinto family. The architectural complex of the factory includes three buildings: the 14th century *alquería de Barrinto*, the *nave* used for production activities and the *nave* used as a warehouse—attached to the latter and probably from a few years earlier, both from the beginning of the 20th century [41].

The peanut oil factory was founded in 1916 and was then redirected in the 1930s to produce olive oil through a steam mill, whose oven and chimney are still visible today and are in a good state of conservation. While the *alquería* has been the subject of a recent restoration in the municipal library Joanot Martorell [41], the factory *naves* (production area and warehouse) will be destined, following the restoration in progress, to hold the activities of the centre for the elderly on the ground floor and an oil museum on the first level. The production activity remained active until 1998, when the municipality expropriated the construction of the Marxalenes Park (Figure 2). The building, disused due to a fire in 2006, suffered the collapse of the original roof in wooden trusses. Since 2016, work has begun to recover the architectural complex of the factory.

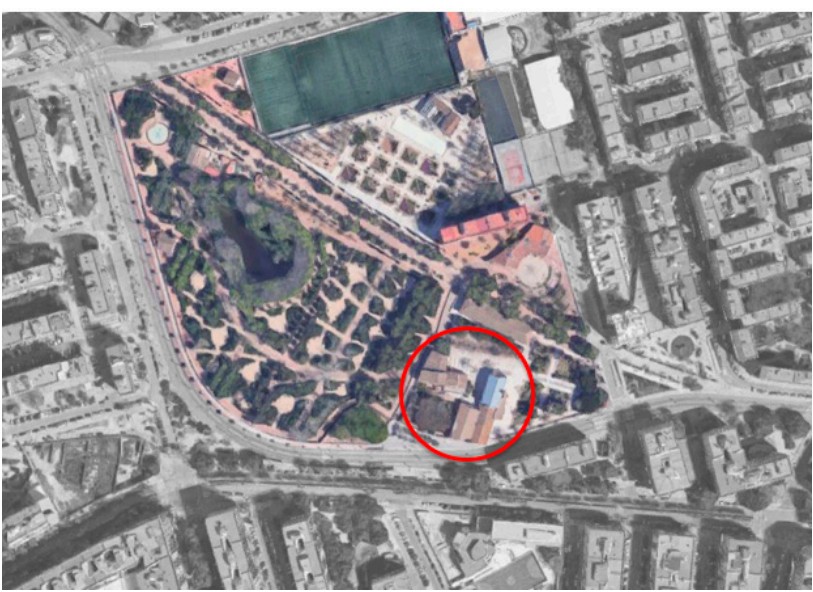

**Figure 2.** The Marxalenes Park. The red circle highlights the localization of the ancient aceitera.

The architectural complex still consists of three independent buildings, but in a functional relationship. This study focuses on the building block containing the oven and the chimney and on the adjacent structure, corresponding to the headquarters of the previous production activities. For half of the longitudinal section, this last functional block is arranged on two levels with independent access from the courtyard (Figure 3).

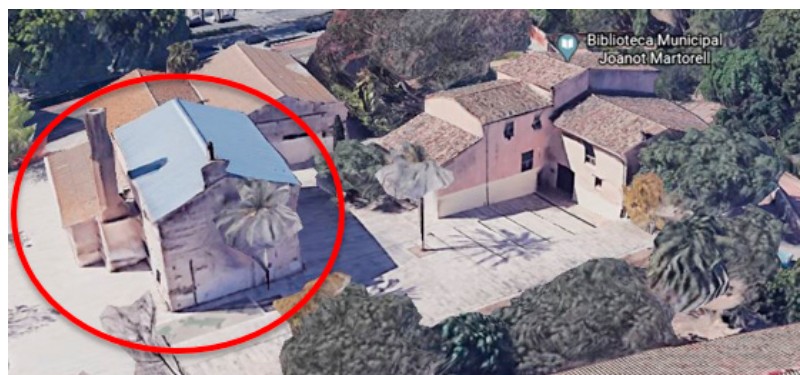

**Figure 3.** An external view of the whole architectural complex (image from Google Maps). The red circle highlights the localization of the two buildings investigated in this work.

Even the two blocks under analysis communicate exclusively through the external distribution space but present a common brick partition on which the octagonal chimney on a quadrangular base stands. The only direct communication between the two environments

occurs through the chimney ground connection: the quadrangular base has a duct (currently buffered) at the ground level that connects the base of the chimney both with the interior and the outside (Figure 4).

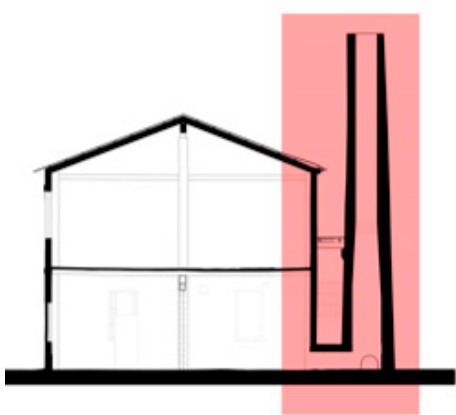 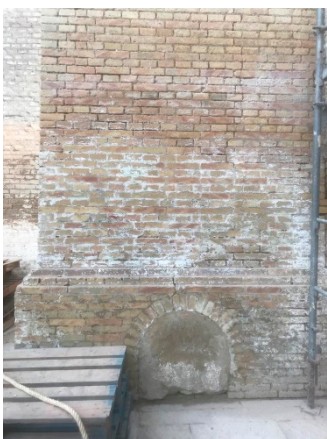

**Figure 4.** Relationship between the chimney, the building and the external space. On the **left**: a transversal section of the buildings analyzed; on the **right**: a view of the buffered opening on the chimney base.

Thanks to this pre-existing duct, it was possible to imagine an air intake for the rooms and distribute the air flows in relation to both the overpressure imposed by the prevailing winds and the delta of temperatures and pressures existing between the various communicating environments.

Therefore, the activation of passive cooling and natural ventilation involved, firstly, the restoration of the buffered duct—with the relative re-proposal of the buffered openings (one outside and one inside)—secondly, the insertion of a chimney pot with a t-section at the top of the ventilation duct and oriented in the direction of the prevailing winds to optimize the draft of the chimney.

The intervention is completed with the preparation of further connections between the chimney and the building block to be ventilated in order to define dedicated routes for the inlet and outlet air flows (Figure 5).

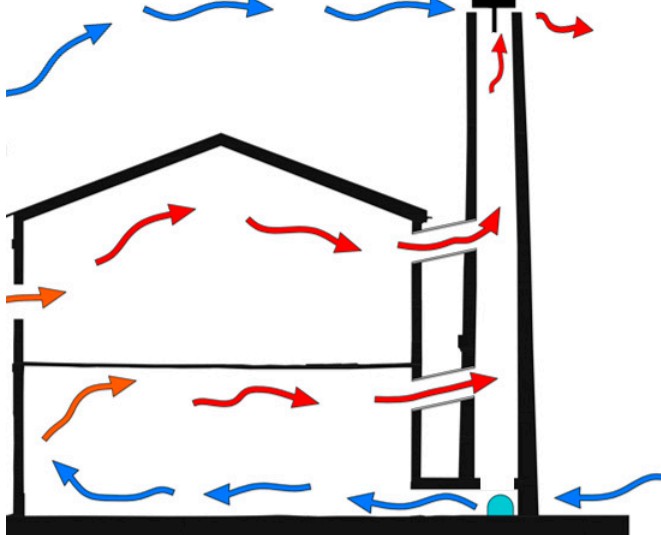

**Figure 5.** Schematic representation of the air flows through the chimney. Red arrows: hot air; blue arrows: cold air.

*2.3. Numerical Methodology*

The analysis of the "Aceitera" ventilation through the dismissed chimney, the task of this work, involves the study of natural convection with heat transfer effects. The simulation of such kinds of processes has been performed relying on the OpenFOAM library software and, in particular, on the *buoyantPimpleFoam* solver. It uses a computational fluid dynamic (CFD) approach featuring the finite volume method and, in particular, the PIMPLE algorithm for solving the governing equations of buoyant, turbulent flow of compressible fluids in transitory ventilation and heat-transfer problems. The PIMPLE algorithm is a combination of PISO (pressure implicit with splitting of operator) and SIMPLE (semi-implicit method for pressure-linked equations) [43].

The mass conservation (continuity) equation of *buoyantPimpleFoam* is given by the following equation [44]:

$$\frac{\partial \rho}{\partial t} + \nabla \cdot (\rho u) = 0 \tag{1}$$

While, in the presence of gravity body force (buoyancy), the momentum conservation equation is [42]:

$$\frac{\partial(\rho u)}{\partial t} + \nabla \cdot (\rho u u) = \& - \nabla p + \rho g + \nabla \cdot \left(2\mu_{eff}D(u)\right) - \nabla\left(\frac{2}{3}\mu_{eff}(\nabla \cdot u)\right) \tag{2}$$

where $u$ is the velocity field, $p$ is the pressure field, $\rho$ is the density field, and $g$ is the gravitational acceleration. The effective viscosity $\mu_{eff}$ is the sum of the molecular and turbulent viscosity and the rate of strain tensor $(u)$ is defined as $(u) = \frac{1}{2}(\nabla u + (\nabla u))$.

In terms of the implementation in OpenFOAM, the pressure gradient and gravity force terms are rearranged in the following form:

$$\begin{aligned} -\nabla p + \rho g &= -\nabla\left(p_{rgh} + \rho g \cdot r\right) + \rho g \\ &= -\nabla p_{rgh} - (g \cdot r)\nabla \rho - \rho g + \rho g \\ &= -\nabla p_{rgh} - (g \cdot r)\nabla \rho \end{aligned} \tag{3}$$

where *prgh* = $p - \rho g \cdot r$ and $r$ is the position vector.

The energy equation is solved for sensible enthalpy [42]:

$$\frac{\partial(\rho h)}{\partial t} + \nabla \cdot (\rho u h) + \frac{\partial(\rho K)}{\partial t} + \nabla \cdot (\rho u K) - \frac{\partial p}{\partial t} = \nabla \cdot \left(\alpha_{eff}\nabla h\right) + \rho u \cdot g \tag{4}$$

where $K = |u|^2/2$ is kinetic energy per unit mass and the enthalpy per unit mass, $h$ is the sum of the internal energy per unit mass $e$ and the kinematic pressure ($h = e + p/\rho$).

The effective thermal diffusivity $\alpha_{eff}$ is the sum of laminar and turbulent thermal diffusivities:

$$\alpha_{eff} = \frac{\rho \nu_t}{P_{r_t}} + \frac{\mu}{P_r} = \frac{\rho \nu_t}{P_{r_t}} + \frac{k}{c_p} \tag{5}$$

where $k$ is the thermal conductivity, $c_p$ is the specific heat at constant pressure, $\mu$ is the dynamic viscosity, $\nu_t$ is the turbulent (kinematic) viscosity, $P_r$ is the Prandtl number and $P_{r_t}$ is the turbulent Prandtl number.

The reliability of the numerical approach just exposed has been assessed using experimental particle image velocimetry (PIV) data measured from a scaled ventilation tower. Full details and validation results are reported in Appendix A. Next, once having shown the validity of *buoyantPimpleFoam* for the simulation of ventilation problems, the code was applied to the Aceitera case study.

*2.4. Cases Study Setup*

The computational fluid dynamic (CFD) simulations aim to demonstrate the effective possibility of performing air ventilation of the Aceitera building by exploiting the dismissed chimney. At the state-of-the-art, the chimney unit is disconnected from the main space.

Consequently, the air quality is bad in terms of humidity, temperature values and ventilation because there is no way to perform air exchange within the main room. The idea is to connect two such spaces through ad hoc ventilation ducts, transforming the chimney into a ventilation system. The developed model allows one to evaluate the thermo-fluid-dynamic behaviour of the system and the chimney draught in different seasons. The proposed solution exploits the wind through the chimney, using its energetic content to promote air exchange inside the building. The chimney has been divided into two parts: the wind entrance from the external and the outlet, for the air exit as shown in Figures 6 and 7. The top of the structure features a wind tower for capturing the air, especially in the summer season. Furthermore, another available air intake at the bottom of the chimney will be used for winter ventilation, as explained in the following.

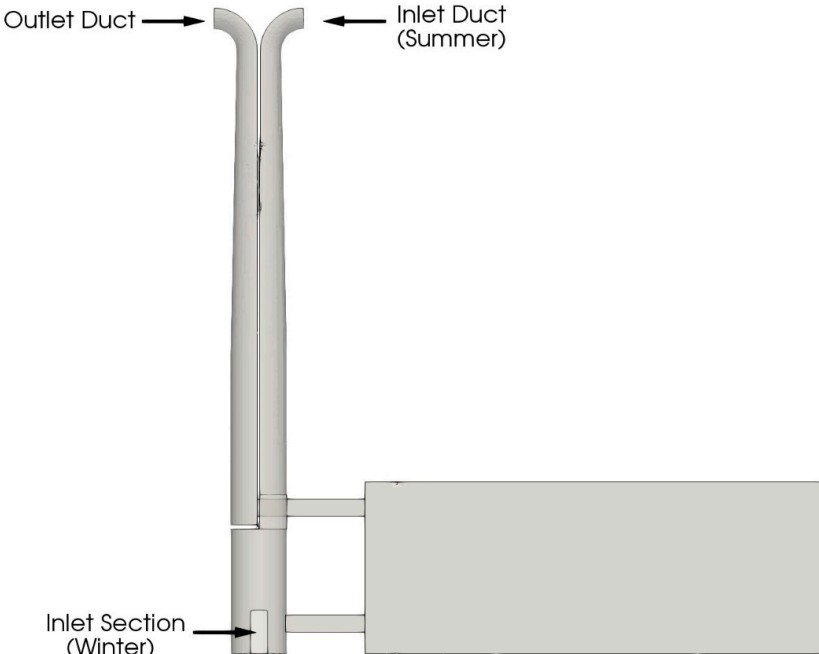

**Figure 6.** Overview of the geometry developed considered for the CFD investigations (original material).

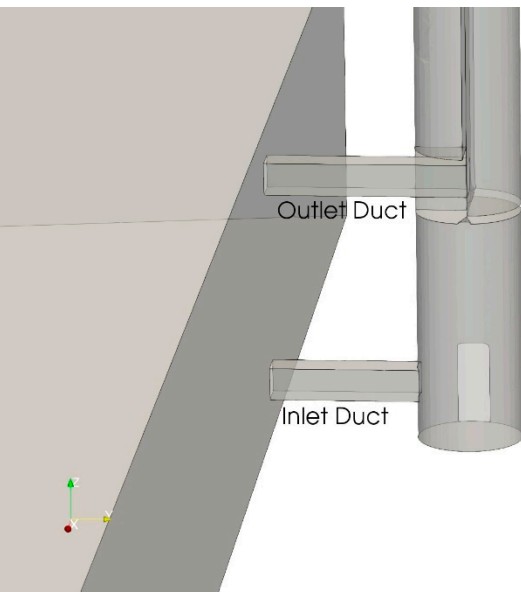

**Figure 7.** A particular view of Inlet and Outlet ducts (original material).

Two operating configurations have been considered: winter and summer season ventilation.

- Winter ventilation: in this configuration, air spontaneously enters from the bottom entrance and exits from the build due to the pressure gradient caused by buoyancy. Indeed, the air inside the chimney, hotter than the external, has higher pressure with respect to colder external air. In this way, air moving from high-to-low pressures goes towards the exit section at the top of the chimney, promoting chimney draught. In other words, fresh air enters the lower part from the restored opening at the chimney basement and is driven into the primary environment by the lower duct. Then, air exits through the upper duct that connects the main room to the upper part of the chimney.
- Summer ventilation: in this situation, the external air is hotter and has a higher pressure than the colder air present within the build. Furthermore, the wind blowing forces air to enter the intake duct, this time, through the wind tower placed on the top of the chimney. In this way, the external air reaches the main environment, then, thanks to the outlet duct, the air is free to go outside realizing the desired clean-up. In this case, the inlet section at the bottom of the chimney is considered closed.

Exploiting the available architectural drawings, a CAD model was used for the CFD simulations. The computational domain features the ground floor of the main building and, obviously, the chimney with the developed connection ducts. The geometry was simplified to be suitable for the *snappyHexMesh* meshing tool. Two unstructured grids were created to perform a grid sensitivity analysis. The base dimensions are of 0.5 m × 0.5 m × 0.5 m. A refinement region has been set for the chimney zone up to a mesh of dimension 0.12 m × 0.12 m × 0.12 m for the coarse grid, while up to 0.06 m × 0.06 m × 0.06 m for the fine grid. These values guarantee solution accuracy and, in the meantime, a reasonable solution timing limiting the total cell count to 300k cells. In both ventilation conditions, the kind of boundary conditions are, essentially, the *Inlet* section for the air entrance in the chimney, *Walls* representing the real building wall the *Outlet* at the chimney top. Tables 1 and 2 report the boundary conditions for winter and summer cases with fixed temperature and velocity values for the inflows. For both the cases and the respective inlet sections, the velocity vector components were fixed accordingly with the wind characteristics described in the next. The air temperature is fixed accordingly with statistical meteorological data of Valencia. The building walls feature a "slip" condition that does not expect a boundary layer formation. The "ZeroGradient" condition specifies a zero equal gradient value on the patch where it is applied. The boundary condition "*fixedFluxPressure*" is used for pressure in situations where a null gradient is generally used (Neumann condition), but body forces such as gravity are present in the solution equations. The condition adjusts the gradient accordingly. The "*inletOutlet*" boundary condition sets the patch value to a user-specified fixed value for reverse flow while the outflow is treated using a zero gradient (Neumann) condition.

**Table 1.** Winter case simulation boundary conditions (original material).

| Field | Velocity u (m/s) | Static Pressure p_ρgh (Pa) | Temperature T (°C) |
| --- | --- | --- | --- |
| Inlet Duct (Top-Summer) | *slip* | *fixedFluxPressure* | *Zero Gradient* |
| Inlet Section (Bottom-Winter) | **(1.12, −0.42, 0)** | *fixedFluxPressure* | **12** |
| Build Walls | *slip* | *fixedFluxPressure* | *Zero Gradient* |
| Outlet Duct | *Zero Gradient* | *fixedFluxPressure* | *inletOutlet* **12** |

**Table 2.** Summer case simulation boundary conditions (original material).

| Field | Velocity u (m/s) | Static Pressure p_ρgh (Pa) | Temperature T (°C) |
|---|---|---|---|
| Inlet Duct (Top-Summer) | **(0, 3.75, 0)** | *fixedFluxPressure* | **28** |
| Inlet Section (Bottom-Winter) | *slip* | *fixedFluxPressure* | *Zero Gradient* |
| Build Walls | *slip* | *fixedFluxPressure* | *Zero Gradient* |
| Outlet Duct | *Zero Gradient* | *fixedFluxPressure* | *inletOutlet* 28 |

In this case the computational domain has been initialized with:

- p_ρgh = 100,000 Pa
- u = 0 m/s
- T = 18 °C

In this case the computational domain has been initialized with:

- p_rgh = 100,000 Pa
- u = 0 m/s
- T = 22 °C

The wind speed has been estimated from specific measurements performed in-site and in the following reported (Figure 8).

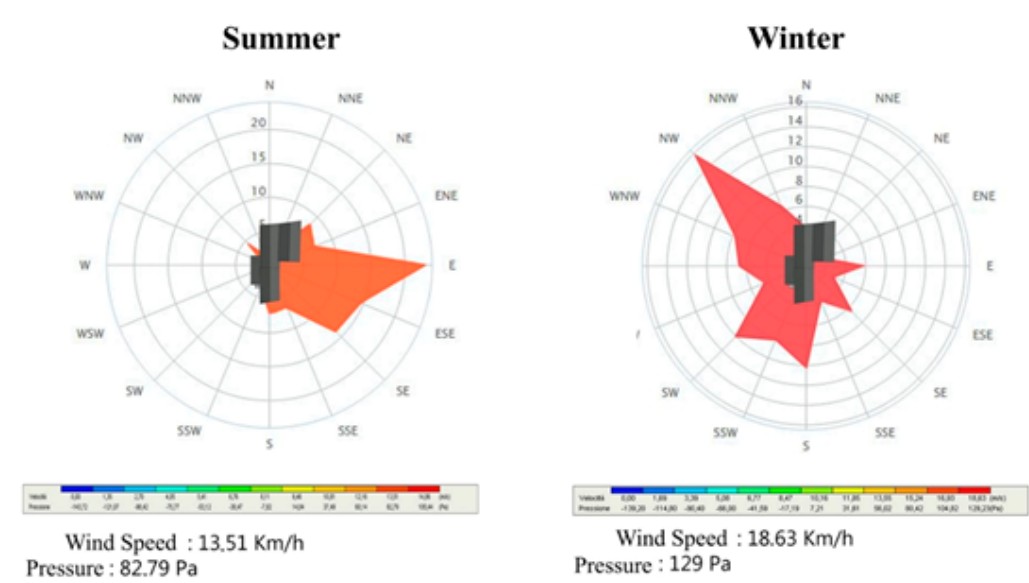

**Figure 8.** The wind rose for the Aceiteira building.

Figure 8 reports the direction and magnitude of wind for the winter and summer seasons, it can be exploited for setting the simulations inlet boundary conditions. Besides, ground-level obstacles, such as vegetation, buildings, and topographic features, tend to slow the wind near the surface. In order to take into account such effects, the wind speed has been considered undisturbed at the chimney top and reduced going down to the ground. The decrease of wind speed with height in the lowest 100 m can be described by this logarithmic expression [45]:

$$u \approx u_{ref} \cdot \frac{ln\left(\frac{z}{z_0}\right)}{ln\left(\frac{z_{ref}}{z_0}\right)}$$

where:

- $u$ = velocity to be calculated at height $z$
- $z$ = height above ground level for velocity $u$
- $u_{ref}$ = 5 m/s undisturbed velocity at height $z_{ref}$
- $z_{ref}$ = 14 m reference height, in this case considered the chimney height
- $z_0$ = 0.4 roughness length in the current wind direction

Applying this relation allows calculating speed at the bottom of the chimney, resulting, for winter period, equal to:

$$z = 0.5 \, \text{m} \quad u = 1.12 \, \text{m/s}$$

The simulated period is of two hours. A variable time-step has been chosen, ruled by a maximum Courant number equal to 2. A RANS Standard k-ε model was used for turbulence simulation. It features two extra transport equations to describe the turbulent properties of the flow. This allows a two-equation model to account for effects such as convection and diffusion of turbulent energy. The simulations have been performed on a Fujitsu Siemens workstation equipped with two Intel Xeon Gold 6140. Numerical results have been post-processed by Paraview software.

## 3. Results

This section reports the simulation results to evaluate the proposed solution's effective capability in performing the "Aceiteira" building air exchange. Following Table 3 reports the average air speed within the main room and mass flow entering/exiting the system after one hour of the air-exchange process.

**Table 3.** Average values of air speed and mass-flow rate after one hour of air exchange.

| Ventilation Case | Mesh Size | Average Air-Speed (m/s) | Inlet Mass-Flow (kg/s) | Outlet Mass-Flow (kg/s) |
|---|---|---|---|---|
| Winter | Coarse | 0.112 | 0.565 | 0.573 |
|  | Fine | 0.133 | 0.549 | 0.543 |
| Summer | Coarse | 0.336 | 1.632 | 1.628 |
|  | Fine | 0.473 | 1.678 | 1.697 |

Minor differences can be observed between the coarse and the fine grid of a few percentage points. The higher wind speed present in the summer condition brings higher values of both mass-flow rate and average air speed within the building. The proposed chimney connection allows a continuous air stream within the "Aceiteira", as demonstrated by the equality of inlet and outlet mass-flows.

Figures 9 and 10 show a particular view of the chimney zone after one hour of air-exchange, highlighting the airflow direction within the unit in the winter and summer seasons. In the first case, the arrows indicate that the air enters from the bottom opening and quickly flows towards the main room. The air stream in the upper ducts is directed from the room to the chimney and, thus, outwardly as desired.

Winter Ventilation Case

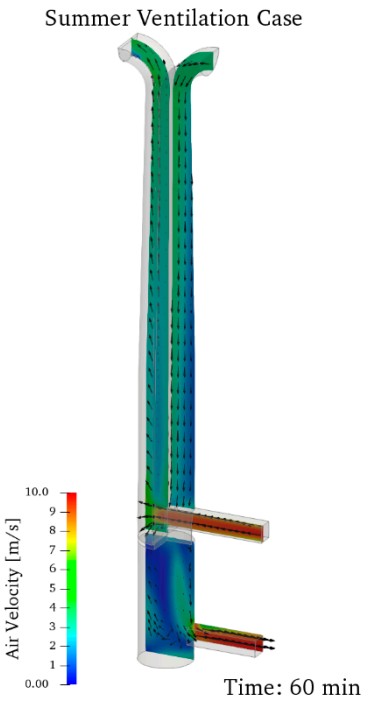

**Figure 9.** Winter ventilation case, zoom in the chimney zone. The air enters from the bottom entrance, flows towards the main space, and exits through the upper duct.

Summer Ventilation Case

**Figure 10.** Summer ventilation case, zoom in on the chimney zone. The air enters from the top right duct and exits from the other duct of the wind tower.

In the summer ventilation case, the air enters from the top right duct exposed to the wind and flows towards the main space, then exits through the other duct of the wind tower in the same way as the winter situation performing the desired ventilation.

The following pictures (Figures 11 and 12) describe the temporal evolution of both temperature and velocity of the air. As can be observed from Figure 11, the temperature within the building decreases continuously due to the effect of the colder external air that enters into the main building. Such behaviour demonstrates the effective capability of the

proposed solution in performing air exchange. The air speed assumes values around one m/s near the chimney connection ducts while, farther, it is one order of magnitude lower.

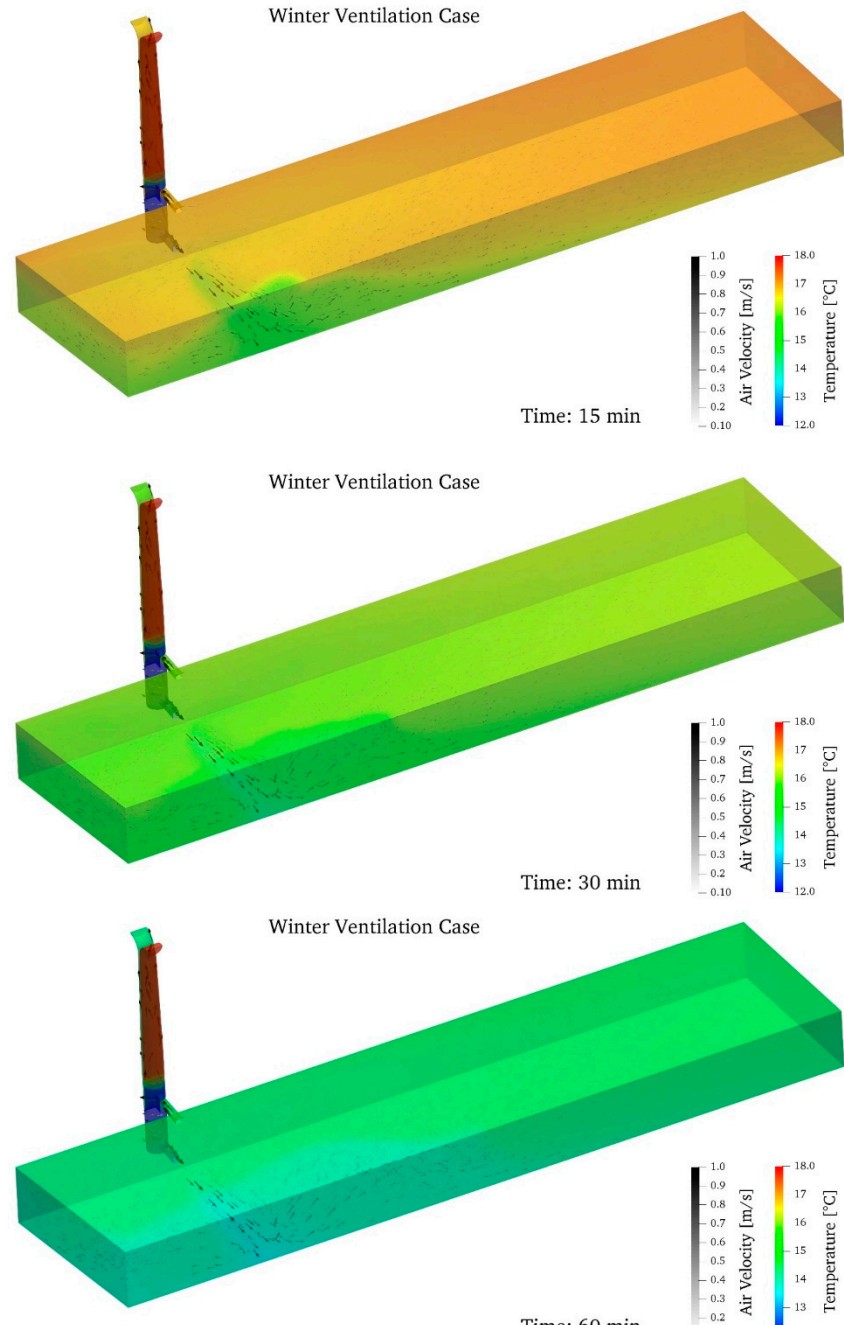

**Figure 11.** Temporal evolution of fluid-dynamics conditions within the "Aceitera" for the winter case.

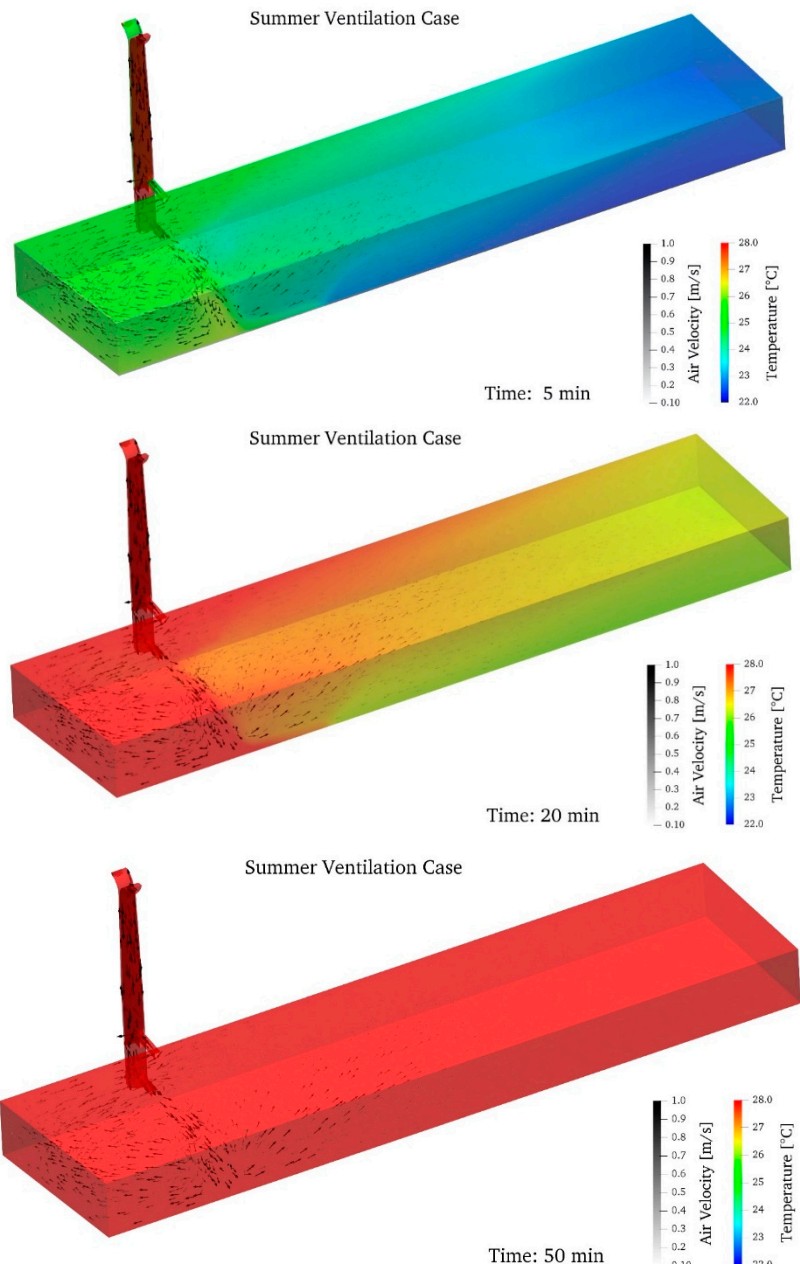

**Figure 12.** Temporal evolution of fluid-dynamics conditions within the "Aceitera" for the summer case.

A faster air-exchange process takes place in summer ventilation conditions (Figure 12). Indeed, the higher airspeed promotes a quicker clean-up, and, in one hour, the external temperature is achieved. It must be noted that, also in this condition, the part of the room closer to the chimney ducts presents higher values of air speed, and so, air-exchange begins here, while farther air speed is one order of magnitude lower.

## 4. Discussion

The following graphs of Figure 13 report the average temperature within the main room plotted against time.

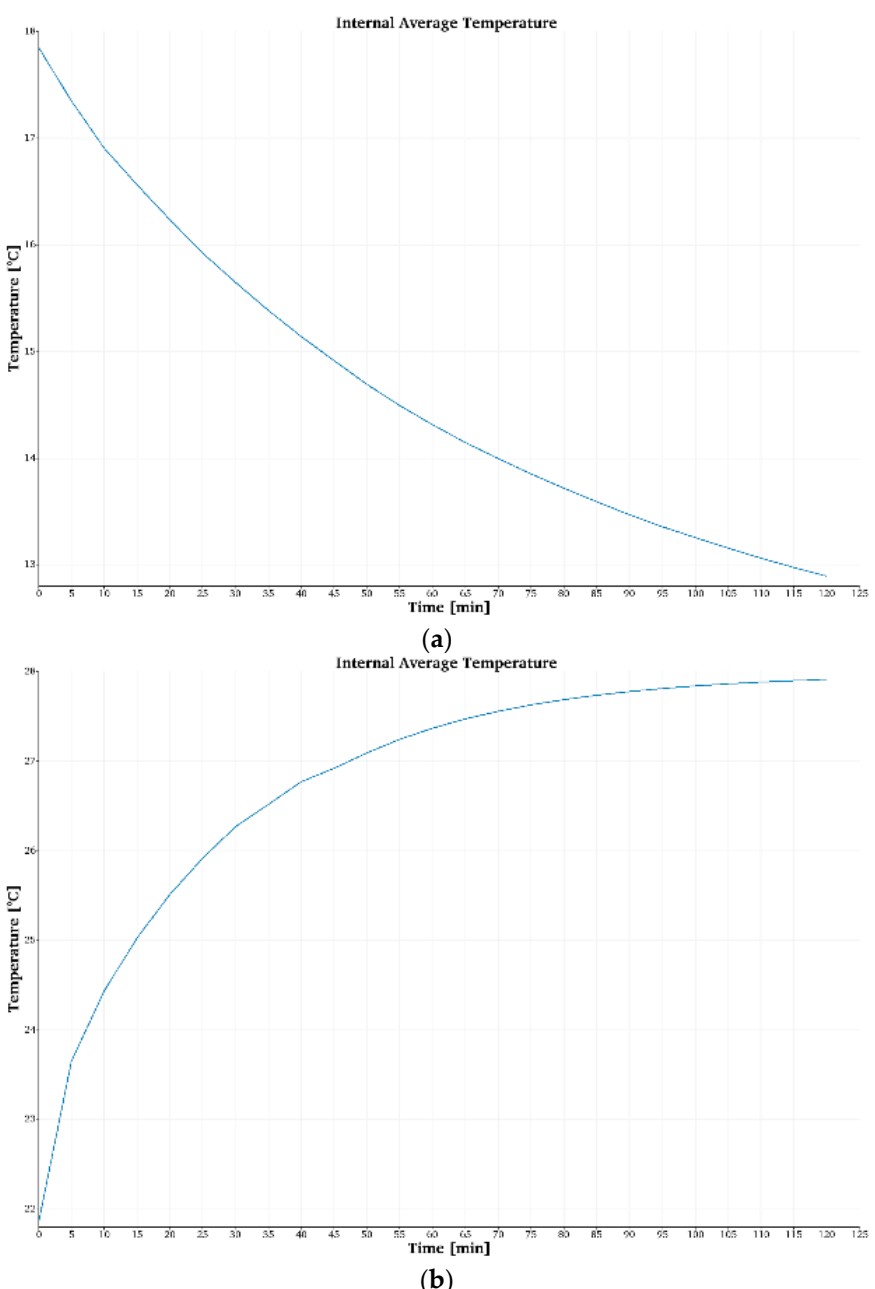

**Figure 13.** Average temperature within the main space of "Aceiteira" building in winter (**a**) and summer (**b**) conditions.

Observing the data, it can be asserted that, after two hours of ventilation in winter and one hour in summer, the average temperature within the main room is almost equal to the external environment temperature. This behaviour, on one hand proves the proficient air-exchange achieved through the passive source, and, on the other, highlights the need for controlling the intake airflow in order to avoid internal uncomfortable temperatures.

However, accurate thermal comfort prediction cannot rely exclusively on air temperature and air speed. Predicted mean vote (PMV) is the most widely used approach; it has been adopted by international standards, such as ASHARE 55 and ISO 7730 [46]. It accounts for two occupants' parameters (metabolic rate and clothing insulation) and four environmental parameters (air temperature, air velocity, mean radiant temperature, and relative humidity). The thermal sensation of a human being is mainly related to the body's thermal balance as a whole. This balance is influenced by physical activity and clothing and

the environmental parameters: air temperature, mean radiant temperature, air velocity, and air humidity. When these factors have been estimated or measured, the thermal sensation felt by a human being can be predicted by computing the predicted mean vote (PMV).

For the Aceitera case study, simulations from CBE thermal comfort tool have been carried out to predict changes in thermal comfort conditions before and after ventilation interventions [47]. Below, graphs from summer and winter conditions have been reported according to the following input data (see Table 4):

**Table 4.** Input parameters form ISO 7703.

|  | Air Temperature (°C) | Mean Radiant Temperature (°C) | Relative Humidity (%) | Average Air-Speed (m/s) | Metabolic Rate | Clothing Level |
|---|---|---|---|---|---|---|
| Summer pre-intervention | 28 | 22 | 90 | 0 | 1.7 | 0.5 |
| Summer post ventilation activation (60 min) | 28 | 26.4 | 68 | 0.47 | 1.7 | 0.5 |
| Winter pre-intervention | 12 | 18 | 50 | 0 | 1.7 | 1 |
| Winter post ventilation activation (60 min) | 12 | 16.4 | 55 | 0.13 | 1.7 | 1 |

Input data were derived from the results of the CFD simulations (airspeed and air temperature), from the online comfort tool (radiant temperature, metabolic rate and clothing level) and from the use of the Mollier chart (relative humidity).

The graphs in Figure 14 show how, in summer, the activation of passive ventilation leads the structure to have conditions of thermohygrometric comfort: observing the summer ventilation condition, it can be inferred that, after ventilation, the optimal thermal comfort was achieved and maintained during the operation of the duct (red circle falls into the light blue comfort zone). Figure 15 shows a different situation for winter conditions: before the ventilation begins, the main environment, considered at 18 °C, is already in a quasi-optimized condition; the entrance of cold air from the outside cools down the environment, bringing the PMV to large negative values: in winter (Figure 15) the discomfort is not resolved by the activation of the ventilation, but worsens slightly due to the lowering of the internal temperature and the consequent increase in relative humidity.

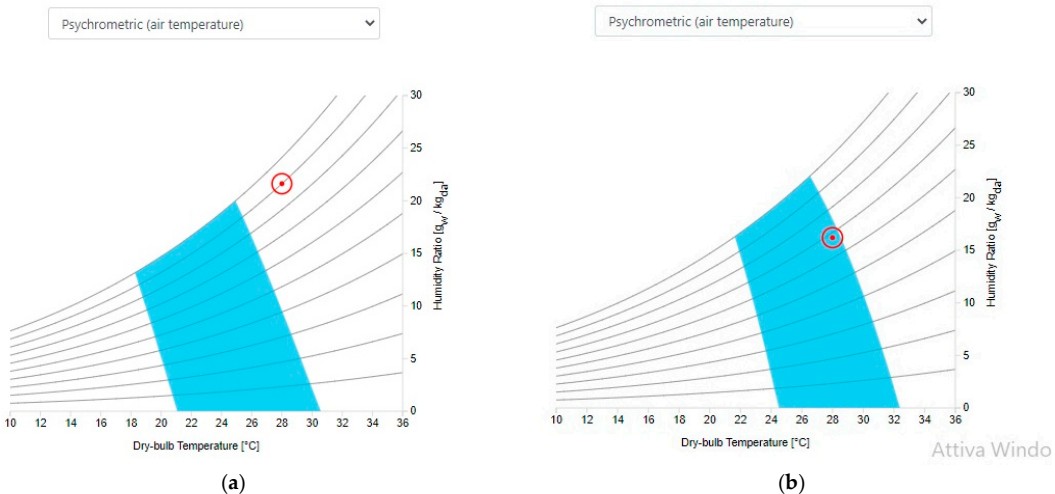

(**a**)          (**b**)

**Figure 14.** Comparison between summer comfort conditions in the Aceitera (the light blue area defines the comfort zone on the chart depending on thermoygrometic parameters; the red circle shows the actual indoor comfort conditions): (**a**) pre-intervention conditions; (**b**) conditions after activation of passive ventilation (after 60 min).

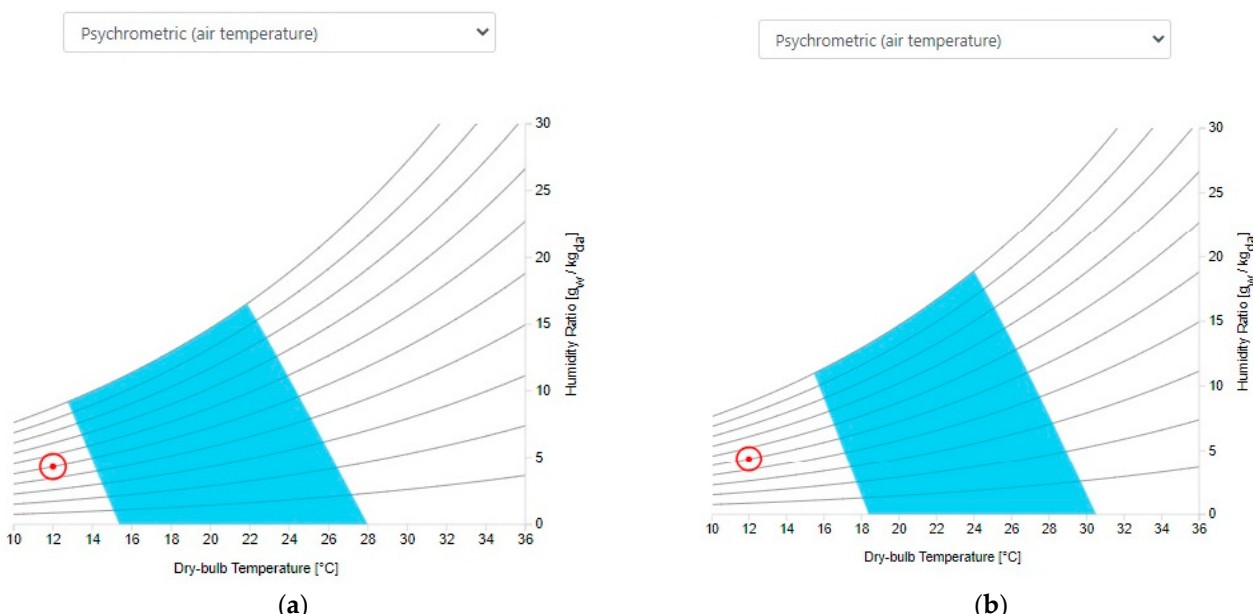

**Figure 15.** Comparison between winter comfort conditions in the Aceitera (the light blue area defines the comfort zone on the chart depending on thermoygrometic parameters; the red circle shows the actual indoor comfort conditions): (**a**) pre-intervention conditions; (**b**) conditions after activation of passive ventilation (after 60 min).

Indeed, the wind speed values present in both summer and winter bring excessive ventilation for desired purposes if left uncontrolled. Such behaviour highlights the unavoidable necessity of heating the external air when ventilation is performed during cold seasons. It follows that mechanical systems for opening/closing the chimney inlet and outlet ducts, eventually retro-controlled by internal temperature sensors, must be installed. The possibility of managing the incoming and outgoing airflow would also make it possible to optimize the use of the ventilation chimney according to the external climatic conditions and the needs of users (crowding of the premises, fruition mode, people metabolic activity, etc.).

From an architectural point of view, the openings for air circulation could be modular to regulate the air flow and connected with the sensors network-for detecting the internal and external thermohygrometric conditions that change according to the weather and the seasons, so that the ventilation duct is made operational to achieve a good level of comfort in the environment in several configurations. It is highlighted how a device designed in this way is capable of modifying its functionality by adapting to the surrounding conditions, which lends itself to being used for a reuse of the building that is adaptive and, therefore, variable over time.

In order to make the retrofit adaptive too, the interventions on the casing and chimney must be carried out with minimum invasiveness. In order to activate the airflow, the chimney is connected to the living space through two connection ducts: one at the base and one higher, near the roof level, as explained in Section 2 and verified in Section 3. The lower duct can be easily made by restoring the pre-existing ancient openings on the quadrangular base of the chimney (Figure 4). These, in fact, connect the chimney both with the outside and with the inside of the living space. As regards the higher-placed duct, this must instead be realized, as there are no pre-existing openings on the masonry. This operation presents a certain degree of invasiveness, which can, in any case, be controlled through works that guarantee the recognition of the intervention and the possibility of restoring the ante-operam state. The masonry is made up of a row of bricks arranged in *Aparejo Flamenco*: the new opening should respect the arrangement of the bricks by providing for their controlled and catalogued removal for future reintegration.

Furthermore, the realization of the duct must be carried out with light materials, preferably metal alloys. At the same time, the restoration of the original openings in the masonry for the lower connection duct allow one to recover the building's historical configuration; this work, therefore, allows for an enhancement of the artifact through the highlighting of some prerogatives of its historical functioning connected to the industrial activities related to the chimney and its connections to the internal rooms. As for the t-shaped chimney to be installed at the top of the chimney (see Section 4), the use of light materials (aluminium profiles or metal sheets) and reversible anchors would allow for easy removal without damaging the underlying structures and materials.

## 5. Conclusions

This paper shows the conversion of an ancient chimney belonging to the industrial heritage into a ventilation duct for indoor comfort enhancement of the historic environment. The case study, the Aceitera de Marxalenes, is located in Valencia (Spain), which is characterized by a hot-summer Mediterranean climate for which cooling interventions are appropriate to achieve comfort conditions, particularly in summer. The adaptation of the chimney into a ventilation duct was designed through simple and minimally invasive architectural interventions on the envelope. The interventions, mainly consisting of the reopening of ancient, buffered openings on the quadrangular chimney base, have been identified to be compatible with the structure, respecting the protection of the historical, formal and material values. The interventions described making the ventilation chimney operational allow the restoration of the ante-operam state in any case. Furthermore, the solutions applied to the chimney have good reversibility and the functionality of the device, in future development, could be easily used to improve mechanical activation of the various chimney openings designed. Due to its characteristics, the planned intervention can be contextualized as a passive strategy for improving the performance of a historic building; furthermore, this retrofit strategy can be easily applied to an important slice of industrial heritage—often subject to adaptive reuse—equipped with disused historic chimneys. The device shown in this paper is capable of making the use of the structure and the retrofit itself adaptive. The effectiveness of the device has been tested through CFD simulations in summer and winter regimes. The numerical approach chosen for this work has been previously tested and validated against experimental data concerning a scaled wind tower. The simulation results demonstrate the converted chimney's capability to produce an efficient ventilation of the building. After 60 min of ventilation, the main room-air conditions are:

- T = 26.4 °C, Airspeed = 0.47 m/s in summer ventilation regime.
- T= 18 °C, Airspeed = 0.13 m/s in winter ventilation regime.

The evaluation of the internal comfort demonstrates the necessity of heating the inlet air during the cold season, while, in the summer season, the optimal thermal comfort is achieved. Possible research developments concern quantifying the comfort improvement conditions inside the building thanks to the intervention shown.

**Author Contributions:** M.D.V. designed the research and wrote Sections 1, 2, 2.1, 2.2, 4 and 5, defined the research methodology and performed the comfort analyses; F.D. realized the model, performed the validation and wrote Sections 2.2, 2.3 and 3–5; A.D.V. supervised the simulations and the paper; P.D.B. supervised the research and the paper. All authors have read and agreed to the published version of the manuscript.

**Funding:** This research received no external funding.

**Institutional Review Board Statement:** No applicable.

**Informed Consent Statement:** No applicable.

**Data Availability Statement:** No applicable.

**Acknowledgments:** The authors acknowledge Luis Palmero Iglesias and the Polytechnic University of Valencia (Escuela Técnica Superior de Ingeniería de Edificación) for providing the case study and students of the International Workshop 2020 "La riqualificazione sostenibile del costruito" for having produced images 4 and 5.

**Conflicts of Interest:** The authors declare no conflict of interest.

## Appendix A

The code *buoyantPimpleFoam* has been tested using PIV data of a wind tower. A scaled model of the real building has been placed in a sub-sonic tunnel and the flow field has been measured by means of a laser velocimetry technique (PIV) as reported in [42], where full details of experimental activities and the instrumentations used can be found. The knowledge of these flow distribution details can be used to evaluate the reliability of the numerical approach here chosen.

Figures A1 and A2 represent the geometry of the wind tower studied. The domain has been discretized with a grid whose base dimension is 5 mm in every direction. The grid has been refined up to 0.4 mm within the wind tower zone, as shown in Figure A3.

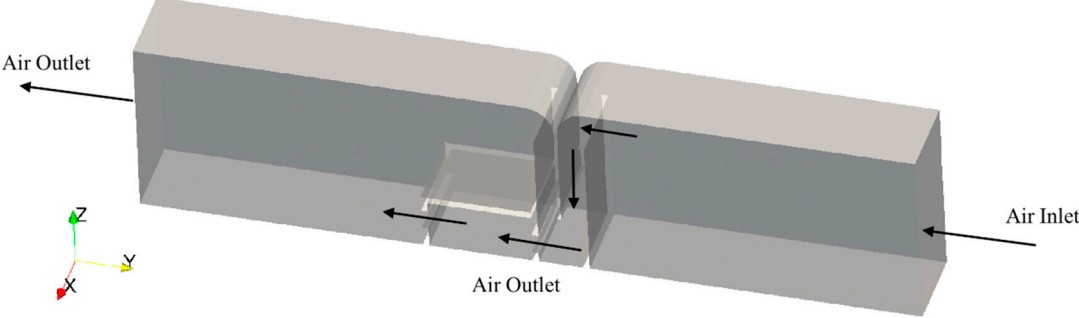

**Figure A1.** Validation model representation.

The air enters from the right side with about 2.8 m/s. It flows towards the main ambient, following the path highlighted by the black arrows in Figure A1.

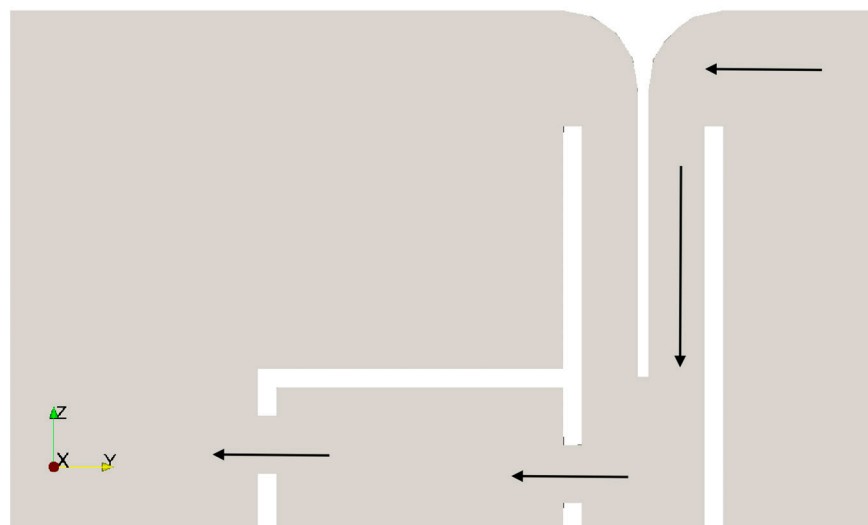

**Figure A2.** Flow path.

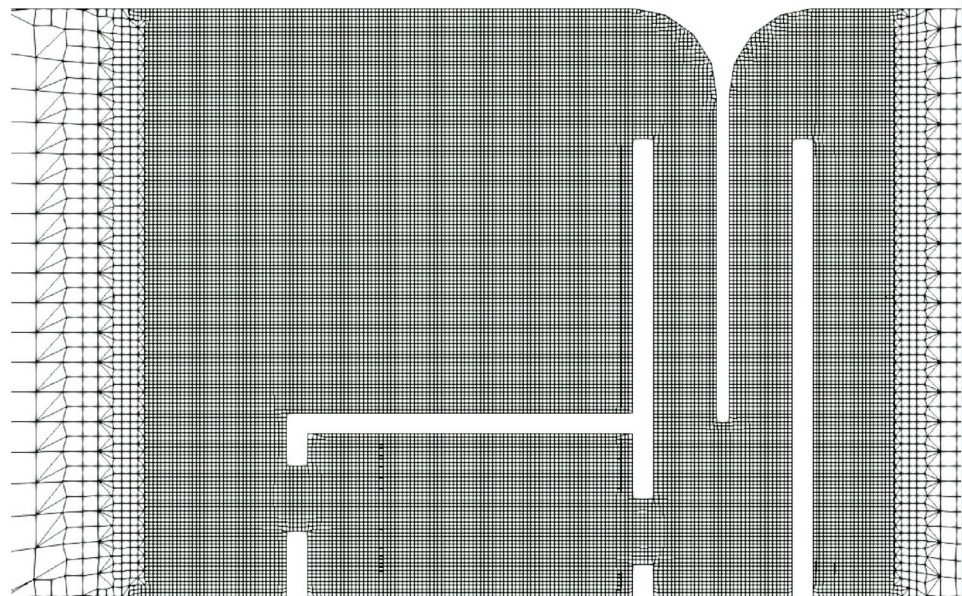

**Figure A3.** A particular view of the computational grid.

Finally, the airflow exits from the building through the bottom opening once overcoming the second room. Fixed outlet pressure of 1 bar has been imposed on the left side of the domain while, at the inflow, the velocity is fixed. Full characteristics of the developed model are reported in Table A1. The boundary condition "*fixedFluxPressure*" is used for pressure in situations where a null gradient is generally used, but body forces, such as gravity, are present in the solution equations. The condition adjusts the gradient accordingly. The "*inletOutlet*" boundary condition sets the patch value to a user-specified fixed value for reverse flow while, the outflow, is treated using a zero gradient (Neumann) condition.

**Table A1.** Validation case boundary conditions.

| Field | Velocity u [m/s] | Static Pressure p_ρgh [Pa] | Temperature T [°C] |
|:---:|:---:|:---:|:---:|
| Air Inlet | (0, −2.8, 0) | *fixedFluxPressure* | **15** |
| Build Walls | *slip* | *fixedFluxPressure* | *Zero Gradient* |
| Outlet | *Zero Gradient* | **101,325** | *inletOutlet* **15** |

A variable timestep, ruled by a maximum Courant number equal to 2, has been chosen. A RANS Standard k-ε model has been used for turbulence representation. The simulation stopped when steady-state conditions were reached and, in particular, residuals lower than $10^{-4}$.

CFD predictions and PIV data are compared in Figure A4 where a complete view of the flow field dynamics in the building-wind tower are shown by means of the velocity streamlines computed from the experimental PIV data on the left side and, from the numerical simulation results, on the right side. Arrows are used to indicate the airflow distribution and its main trajectories inside the building-wind tower configuration. As it can be observed, the main characteristics of the flow field are well represented by the CFD simulation and the vortical structures properly captured, in particular the ones that grow within the main chamber.

As further validation, the velocity profiles at different locations have been plotted in Figure A5. They have been identified as dotted lines in Figure A5a. In particular, they are the end of the inlet tower column, the main room entrance, the middle of the main room, and the outlet through the window.

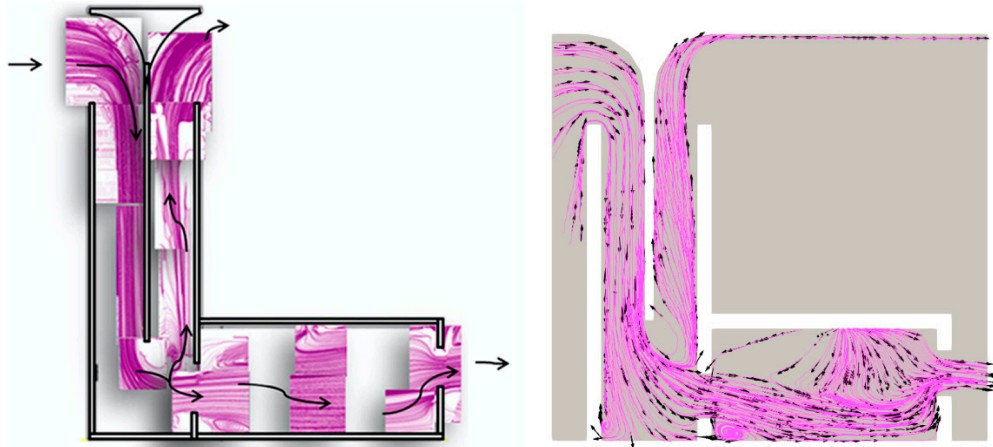

**Figure A4.** Comparison of whole data from PIV measurements and streamlines obtained from CFD simulation.

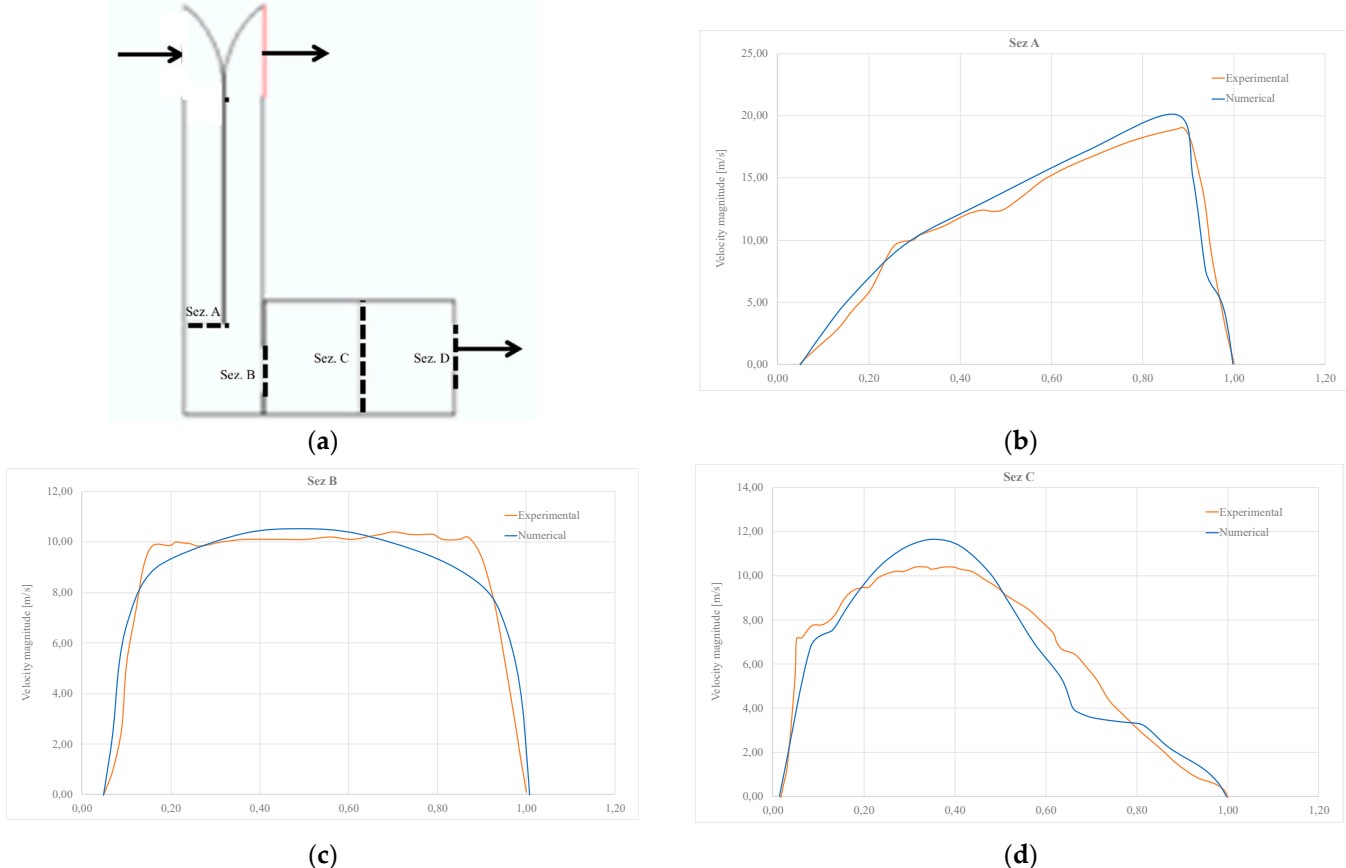

**Figure A5.** *Cont.*

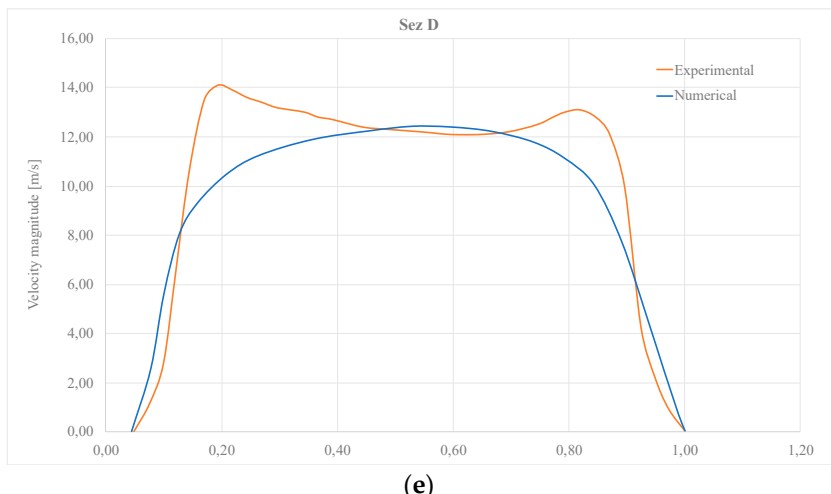

(**e**)

**Figure A5.** Comparison of CFD results and PIV measurement of normalized profiles of velocity across significative sections: (**a**) location for velocity profiles; (**b**) wind-tower end of tower column; (**c**) inlet of main room section; (**d**) main room middle section; (**e**) outlet of main room section.

The simulation results agree with the data obtained from PIV measurements.

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
