# Peer review of "Adaptive Retrofit for Adaptive Reuse: Converting an Industrial Chimney into a Ventilation Duct to Improve Internal Comfort in a Historic Environment"

_sustainability, doi:10.3390/su14063360_

Round 1

Reviewer 1 Report

Review

Converting an industrial chimney into a ventilation duct: activating passive cooling to improve internal comfort in a historic  environment

Authors  Mariangela De Vita , Francesco Duronio  , Angelo De Vita  and Pierluigi De Berardinis ,

Abstract: Please structure the abstract as:

Introduction-Aims

Method

Results and interpretation

Please insert the authors of the equation  …….line 248

The mass conservation (continuity) equation of buoyant Pimple Foam is given by the  following equation

Please insert the authors of the equation  line 251

Idem

The energy equation is solved for sensible enthalpy:    line 261

Fig. 6 to insert the data source

Fig.7 to insert the data source

Table 1 line 329, please insert the data source

Idem Table 2

Fig. 9 and 10 please insert the data source

References

Please cite also:

IlieÅŸ Marin, IlieÅŸ Dorina, Josan Ioana, IlieÅŸ Alexandru, IlieÅŸ Gabriela, (2010), The Gateway of MaramureÅŸ Land. Geostrategical Implications in Space and Time, in Annales. Annals for Istrian and Mediteranian Studies, Series Historia et Sociologia, ISSN 1408-5348, 20, 2010, 2, Zalozba Annales, Koper, Slovenia, pag. 469-480, (http://www.culture.si/en/Annales_Journal);

Marcu, Florin, Ilies, Dorina Camelia, Wendt, Jan A., Indrie, Liliana, Ilies, Alexandru ,  Burta, Ligia Ciaciora, Tudor,  Herman, Grigore Vasile, Todoran, Angela, Baias, Stefan, Albu, Adina, Gozner, Maria, Investigations regarding the biodegradation of the cultural heritage, Case study of traditional embroidered peasant shirt (Maramures, Romania), 2020, ROMANIAN BIOTECHNOLOGICAL LETTER, Volume 25, Issue 2, Page 1362-1368, DOI 10.25083/rbl/25.2/1362.1368

Ilieș D.C., Oneț A., Herman G., Indrie L., Ilieș A., Burtă L., Gaceu O., Marcu F., Baias Ș., Caciora T., Marcu A.P., Oană I., Costea M., Ilieș M., Wendt A.J., Mihincău D., (2019), Exploring indoor environment of heritage buildings and its role in the conservation of valuable objects, in Environmental Engineering and Management Journal, 18(12), 2579-2586, (http://www.eemj.icpm.tuiasi.ro/accepted.htm).

Author Response

Dear Reviewer, 

thank you for your comments. The paper has been revised and improved following your suggestions. Find attached the summary.

Kind Regards, 

The Authors

Reviewer 2 Report

Dear Authors, the well witten paper shows the following criticalities that require more refinement:

  1. Although You declare that the paper stems from an in-depth analysis of the industrial architecture heritage of Valencia (Spain), carried out thanks to the Short Term Mobility research funding from the Italian National Research Council for the years 2018 and 2019 ... not it is clear whether this architecture is unique or whether Your interest is linked to the replicability of observations and proposals.
  2. In the Introduction, You refer to Further technological solutions, line 82, to improve performance levels. The reversibility requirement that appears in the conclusions should already be made explicit here as a hypothesis alongside that of the functional improvement that is supported by the calculation model.
  3. In the Discussion please, it would be useful:
  • Specify the requirement of minimum invasiveness, introduced in line 477 referring to the architectural works and reversibility of the project line 495 that could be necessary to optimize the thermal behavior of the system.
  • Describe the architectural works more extensively, as was stated in line 122 (historic artefacts by exploiting construction elements already integrated into the building). It could be interesting to have information on the compatibility between pre-existing materials and geometries and new technologies.
  • Better justify the hypothesis introduced in the conclusions to line 496 of improving the functionality of the device through the mechanical activation of the various chimney openings designed.
  1. In the Conclusions, the concept of reversibility referred to the proposed solutions is introduced, in line 495. However, in the absence of a detailed description of the architectural works, this concept is not validated. It would be appropriate to clarify the concept underlying this declaration better because the creation of openings in a brick building system, with a geometry like the chimney, will require sewing and untwisting and the use of special pieces, new materials. Therefore, on a constructive level, perhaps the intervention cannot be considered reversible. There is no reflection on the replicability of the proposed study, both in the introduction and in the discussion, which would make the analysis of the case study much more interesting.

Finally, two editorial comments:

  • The instructions for the preparation of the paper have been left on line 483, please check it.
  • Line 182 I think is a title: The case study of the Aceitera de Marxalenes: the activation of the ventilation chimney

Author Response

(The authors gave the same response as above.)

Reviewer 3 Report

This manuscript presented an application study by retrofitting an industrial chimney. This is an interesting topic, however, the structure, method, abstract and results are not sufficiently documented. A major review is required to enhance the quality of this article. Please pay attention to the method and results sections. The description of relevant parts has to be presented, e.g., the grid, boundary conditions, quantitative results.

(1) In the abstract, the contribution including the method and results were not presented in details. There are too many description related to the background and reason of this study.

(2) To reviewer’s knowledge, the literature review is not sufficient and lack some in the recent publications.

(3) What is the PIMPLE algorithm? Please check.

(4) Please provide the mesh info, including the grid independent analysis, cells number.

In the method section, the evaluation indices are lost.

(5) This is an unsteady simulation, please provide the sufficient boundary condition descriptions.

(6) Figure 13 is too large, please reduce it and combine figure 13a and figure 13b.

(7) The description of figure 14 is lost, what is this picture used to say?

(8) The results section is suggested to have the subsections and clearly presents main results with several titles.

(9) In the conclusion and results, please provide a certain quantitative descriptions related to the main findings.

(10) The reference format should be seriously revised according to the journal guideline for authors.

(11) Please ask a native English editor to polish the content.

Author Response

(The authors gave the same response as above.)

Round 2

Reviewer 2 Report

Accept in present form

Reviewer 3 Report

The revised manuscript can be published after the significant revisions.